# A Review of Interactions between Plants and Whitefly-Transmitted Begomoviruses

**DOI:** 10.3390/plants12213677

**Published:** 2023-10-25

**Authors:** Hassan Naveed, Waqar Islam, Muhammad Jafir, Vivian Andoh, Liang Chen, Keping Chen

**Affiliations:** 1School of Food and Biological Engineering, Jiangsu University, Zhenjiang 212013, China; vandoh@ujs.edu.cn; 2School of Life Sciences, Jiangsu University, Zhenjiang 212013, China; oochen@ujs.edu.cn; 3State Key Laboratory of Desert and Oasis Ecology, Xinjiang Institute of Ecology and Geography, Chinese Academy of Sciences, Urumqi 830011, China; ddoapsial@yahoo.com; 4Department of Ecology, School of Resources and Environmental Engineering, Anhui University, Hefei 230601, China; m.jafir@ahu.edu.cn

**Keywords:** insect vector, evolution, virus transmission, interactions, viruses

## Abstract

The transmission of plant viruses from infected to healthy host plants is a process in which insects play a major role, using various transmission strategies. Environmental factors have an impact on the transmission of viruses and the subsequent development of infections or diseases. When viruses are successful, plant virus diseases can reach epidemic proportions. Many plants across different regions are vulnerable to viral infections transmitted by the whitefly vector. Begomoviruses, which are transmitted by whiteflies, represent a significant threat to agriculture worldwide. The review highlights the mechanisms of virus acquisition and transmission by whiteflies and explores the factors influencing these interactions. Understanding the impacts of these changes is crucial for managing the spread of pests and mitigating damage to crops. It underscores the need for continued research to elucidate the mechanisms driving plant–insect–virus interactions and to identify new approaches for sustainable pest management.

## 1. Introduction

According to recent studies, over 80% of plant pathogenic viruses are dependent upon insect vectors for their spread [1,2,3]. Among insects, hemipteran, such as aphids, leafhoppers, plant hoppers, whiteflies, mealy bugs, true bugs, and some treehoppers, dominate as plant virus vectors [4,5].

Among hemipterans, whiteflies are the most important group of vectors. There are more than 1500 species of whiteflies belonging to 161 genera across the world [6,7]. Many plants are known to host various species of whiteflies, which are responsible for transmitting plant viruses. Over the last two decades, research has revealed that whiteflies are the predominant vectors responsible for transmitting plant viruses among insects [2,8]. Whiteflies are polyphagous in nature and have a broad host range. That is why they can effectively spread plant viruses. Based on the association of plant viruses with the vector, plant viruses can be divided into three categories: persistent, semi-persistent, or non-persistent and depending on the virus’s pathway within its vector as non-circulative or circulative. Another classification is included in this area based on the virus retention sites, distinguishing between cuticula-borne and salivary gland-borne viruses. While certain viral and insect proteins have been found to regulate specific virus–vector associations, there is ample potential for further exploration and investigation in this field of research [9]. Among whitefly species, the silverleaf whitefly (*Bemisia tabaci*) cryptic species complex stands out as a competent plant virus vector. Within *B. tabaci*, Middle East–Asia Minor 1 (MEAM1, aka B Biotype/mitotype) and Mediterranean (MED, aka Q biotype/mitotype) are the two most invasive members of the species complex. Both MEAM1 and MED have created havoc on major agricultural production regions of the world, including China, Africa, and the Southern United States [10,11,12]. 

There are over 1100 known types of plant pathogenic viruses, and among these, the family *Geminiviridae* is one of the largest and most important families of plant viruses [13,14,15]. The family *Geminiviridae* is grouped into nine genera, including *Curtovirus*, *Begomovirus*, *Topocuvirus*, *Mastrevirus*, *Eragrovirus*, *Capulavirus*, *Becurtovirus*, *Turncurtovirus*, and *Grablovirus*. Begomoviruses, which constitute the largest genera within *Geminiviridae* are transmitted by *B. tabaci*. The genomic size of begomoviruses (monopartite or bipartite) ranges from 2.7 to 3.0 kb (Figure 1) [16]. The DNA-A of the bipartite begomoviruses comprises six incomplete open reading frames (ORFs), which encode various proteins, including *AC1*/*Rep*, *AC2*/*TrAP*, *AC3*/*REn*, *AV2*, *AV1*/*CP*, and *AC4* proteins [17,18]. All of these proteins are located on both strands (virus sense strand and complementary strand). These encoded proteins are responsible for crucial functions, including virus replication, the encapsidation of genomic molecules, and the regulation of viral gene expression. On the other hand, DNA-B has two ORFs that encode nuclear shuttle proteins (NSPs) and movement proteins (MPs) on the sense strand and complementary strands, respectively [19,20]. DNA-A components can replicate autonomously, while the DNA-B encodes two proteins responsible for facilitating the movement of viral DNA within and between host cells [21]. Alphasatellites, betasatellites, and deltasatellites are distinct types of circular DNA satellites associated with begomoviruses [22]. The intergenic region of DNA-A contains a conserved sequence of about 200 base pairs known as the Common Region (CR), which includes the TAATATTAC sequence v-ori within a conserved stem-loop structure [18].

Among the viruses transmitted by the *B. tabaci*, begomoviruses are particularly noteworthy for their impact on a wide variety of broad-leaf plants and can be effectively transmitted by *B. tabaci* between host plants. Begomovirus transmission by whiteflies is contentious with reports suggesting both circulative non-propagative/propagative transmission, as seen in Table 1 [23,24]. Nevertheless, under both conditions, begomoviruses do circulate in whiteflies. Circulative transmission has also aroused interest in the transovarial transmission of begomoviruses in *B. tabaci*. It also remains unclear with studies reporting both the transovarial transmission and non-transmission of begomoviruses in *B. tabaci* [5,25,26]. When whiteflies feed on infected plants, they pick up the virus particles and carry them in their gut. As they move on to feed on healthy plants, they can transmit the virus to these plants through their saliva. Whiteflies can also transmit several other viruses that also can cause significant damage to different crops (Table 1). Whiteflies are a significant concern for agriculture as they not only spread plant viruses but also cause damage to their host plants by feeding on them [9]. In addition, their nymphs produce honeydew on the lower branches of trees, which attracts sooty mold disease and impacts the quality of leaves and photosynthesis [27,28]. The high infestation of whiteflies in crops can also have a significant negative impact on other factors in the surrounding ecosystem, disrupting natural insect populations and attracting other pests that can harm beneficial insects and pollinators [2,28]. Whiteflies are the primary vector for begomoviruses and can result in substantial economic loss via direct feeding, as well as transmitting them to different crops, including cotton, tomato, cassava, and vegetables [29,30,31]. A majority of the plants are vulnerable to whiteflies transmitting begomoviruses, which mainly infects dicot hosts. This genus includes many species and strains that are known to cause significant damage to crops worldwide, with different transmission rates among various species within the complex [8,30].

Here, we provide an in-depth comprehension of the transmission of begomoviruses to plants through whiteflies. It encompasses various segments that analyze the possible modifications viruses can induce in plant hosts and insect carriers. This evaluation aims to clarify the possible genetic alterations that begomoviruses may undergo in the course of such interactions. Recent research has yielded extensive insights into the biological and molecular aspects of plant–begomovirus transmission facilitated by whiteflies [4,39,40,41,42,43]. When plants come into contact with whitefly vectors in a specific ecosystem, the begomoviruses have the ability to manipulate the behavior of both the host plant and the vector. Few theoretical studies have investigated the genetic changes occurring in these viruses during transmission [4,44]. 

## 2. Whitefly-Mediated Transmission and Acquisition of the Begomovirus

Begomoviruses can trigger cellular and molecular responses in their insect vectors, which can then affect their transmission [45,46]. *B. tabaci* is the most widely distributed among whiteflies, consisting of over 44 morphologically similar species, including the major pest complex in agricultural ecosystems, Middle East-Asia Minor 1 (MEAM 1), SSA1 (Sub-Saharan Africa 1), SSA2 (Sub-Saharan Africa 2), and Mediterranean cryptic species [47,48,49]. Genetically, these cryptic species are divergent and reproductively isolated but cannot be distinguished morphologically [10]. This group of insects not only causes direct feeding damage but also transmits plant pathogenic viruses, with over 90% of the transmitted virus species being the most harmful viruses infecting hundreds of plant species. The complex interactions among the virus, vector, and host plant determine begomovirus transmission, with factors, such as virus acquisition by the associated vector, vector landing and probing, and feeding patterns, all playing a role in the efficiency of transmission [40,45,50]. 

During the transmission of begomoviruses via whiteflies, the viruses undergo various interactions with insect proteins and pass through the barriers in the digestive, hemolymph, and salivary systems before transmission from the insect body [8,51]. The interaction and tissue tropisms within the vector body provide key pathways in determining the efficacy and specificity of transmission [32]. 

Whiteflies transmit persistent viruses through a series of steps that include ingestion, attachment, entry, translocation, circulation, retention, and release of the virus. Whiteflies acquire viral particles while feeding on infected plants. The acquired viral particles (virions) reach into the midgut and esophagus, followed by entry and translocation into the cell across the basal membrane. Following acquisition, replication, and dispersal, they enter into the hemolymph, eventually becoming located in the primary salivary glands. From the salivary glands of the vectors, the viruses are released in saliva and enter into the plant’s phloem. After injection of the virus into the plant via saliva, the virus replicates inside the plant cells and spreads to other parts of the plant, causing leaf yellowing, stunted growth, and a decreased yield [8,32,52]. During this process, viruses encounter various barriers or screenings while interacting with whitefly receptor proteins (Figure 2). This interaction can result in genetic changes, such as mutations or recombinations, leading to the evolution of new viral strains.

Studies with cucurbit leaf crumple virus (CuLCrV) and Tomato yellow leaf curl virus (TYLCV, begomovirus) have shown that the majority of ingested virions are localized in the midgut and filter chamber of the insect vector [53,54]. Some studies have shown a transient increase in TYLCV genomic DNA in the whitefly gut. However, high levels of viral replication might trigger the activation of the insect autophagic response, which causes the degradation of viral particles and limits its spread to other organs, such as the ovaries and fat cells. While some begomoviruses can invade these organs, autophagy acts as a defense mechanism to restrict virus replication and dissemination [8,51]. The movement of begomoviruses in insect vectors is regulated by the structural component AV1 or coat protein (CP). The interaction of several begomoviruses with their insect vectors has been observed to elicit various cellular, molecular, and behavioral responses that could impact vector survival and virus spread [42,55]. Studies have shown that virus-induced changes in vectors at a transcriptional level can affect gene expression patterns related to virus reception, entry, tissue tropism, multiplication, and immune responses [56]. Additionally, the Janus kinase/signal transducer and activator of transcription (JAK/STAT) signaling pathway plays a role in balancing vector fitness and virus transmission, with the virus inhibiting the pathway, while the vector protects itself through it [3]. Furthermore, virus infection can alter vector orientation behavior, settling and feeding behavior, fecundity, and survival, potentially enhancing virus transmission [57,58]. Moreover, the pre-infestation of plants by vector or non-vector insects can affect subsequent viral transmission and infection, with different insect mouthparts activating different plant signaling pathways that impact virus replication and movement [59].

The infection cycle of a begomovirus begins when virus particles are acquired by the whitefly from the plant phloem of the infected plants through the insect’s stylets (Figure 2). Both the acquisition and inoculation access periods for virus transmission may range from 10 to 60 min [60,61,62]. It is noteworthy to recognize that the amount of virus assimilated by different whiteflies can indeed vary, even when they are granted the same level of access to a specific leaf for a consistent duration [63,64,65]. Particularly significant is the trend wherein the frequency of virus transmission increases as the duration of the acquisition access period is extended [66]. However, given the host range of begomovirus and vectors, whiteflies often encounter host plants with varying virus levels, which can profoundly impact the virus transmission. In fact, studies using different pathosystems have shown that begomovirus accumulation in whiteflies follows a density-dependent phenomenon, where the higher the virus accumulation in host plant leaves, the higher the virus accumulation in whiteflies [67,68,69]. Furthermore, an intriguing observation emerges in relation to the potential impact of whitefly gender on the efficiency of virus transmission, wherein male whiteflies exhibit comparatively lower effectiveness as vectors [31].

## 3. Begomovirus-Induced Changes in Whitefly Behavior

Begomoviruses are capable of replicating and building up in the salivary glands of whiteflies, resulting in alterations in their behavior and increased transmission [70,71]. Infected whiteflies tend to exhibit increased probing and feeding behavior on plants, leading to higher transmission rates of the virus. Moreover, studies have shown that whiteflies infected with begomovirus tend to feed for longer periods, thereby increasing virus acquisition and transmission rates between plants. Begomoviruses can have a direct or indirect impact on the behavior, fitness, and life cycle of whitefly insect vectors [31]. The viruliferous whiteflies exhibit reduced mobility and extended periods of salivation while engaging in the consumption of eggplant leaf discs. Changes in whitefly behavior can also occur due to an increase in their density and the accessibility of infected plants in the area [15,40,72]. The behavior of whiteflies, like feeding and preference, can be influenced by viruses and have a significant impact on the spread and severity of viral epidemics [31,72]. Future molecular studies could investigate the complex ways that control these changed behaviors, revealing the specific factors affecting how whiteflies with viruses move and produce saliva. Furthermore, exploring the effects of different factors related to the plants they feed on might help us understand better why more exposure to virus particles leads to higher transmission rates. Looking into these questions could improve our understanding of how vectors and pathogens interact and potentially lead to new methods for managing and controlling diseases.

Some reports suggest that ingestion of phloem sap of viral-infected plants may ultimately affect the fitness of insect vectors and their transmission capability [41,73,74]. For instance, *B. tabaci* increases its feeding on the cucurbit chlorotic yellow virus (CCYV, a non-circulative virus)-infected plants to increase the chance of virus acquisition and subsequent spread [15,38,75]. Canto et al. reported that the physiology of the host plants directly influences the feeding process of the vectors and host selection [76]. Additionally, begomoviruses can modify the phenotypic characteristics of the plants, which may increase virus transmission in heathy plants [77]. Both the circulative and non-circulative viruses can boost vector attraction to the infected host and also improve virus transmission efficiency [45,75,78]. However, the pathways for vector preference modulation are intricate and vary across different virus–plant–insect pathosystems. Numerous studies have shown that viruses can manipulate insects and plants in ways that affect different stages of insect–plant interactions and increase transmission [45,75]. However, the mechanisms underlying vector preference modulation are complex and vary across different virus–plant–insect pathosystems, making it difficult to disentangle them. Previously, studies conducted using the same population of *B. tabaci* MEAM1 under similar conditions indicated that the interactions between the begomovirus, host and vector are pathosystem-specific. For instance, studies with TYLCV-infected tomato demonstrated that *B. tabaci* MEAM1 that acquired no virus was attracted towards susceptible genotypes with higher TYLCV accumulation, and they accumulated higher TYLCV compared with whiteflies feeding on resistant hosts with reduced TYLCV accumulation [31,69]. Also, the *B. tabaci* MEAM1 developmental time decreased significantly on TYLCV-infected susceptible tomato plants compared with non-infected plants. In contrast, in another study, the same *B. tabaci* MEAM1 population that acquired no virus, avoided settling on squash infected with CuLCrV, and whitefly development on CuLCrV-infected squash did not result in any fitness benefits [66].

Plant viruses frequently manipulate the insect vector’s biology, which ultimately impacts virus epidemiology [79,80]. Moreover, these viruses can also modify the production and activity of phytohormones, which affect the abundance of insect vectors. For instance, infection with begomoviruses has been observed to inhibit the biosynthesis/catabolism of jasmonic acid (JA), along with its signaling pathways. This phenomenon contributes to the improved performance of the whitefly vectors associated with the infected plants. In the case of tomato plants, begomovirus infection elevates the biosynthesis/catabolism of salicylic acid (SA), along with its signaling pathways, while simultaneously downregulating JA-related processes. Additionally, terpenoid biosynthesis and catabolism are reduced in begomovirus-infected plants, making the host plants more palatable to whiteflies [39,81]. Zhao et al. [31,69] highlighted how begomoviruses can manipulate plant immunity to not only enhance the fitness of their whitefly vectors but also suppress the performance of non-vector insects. In the context of tobacco infected with the tomato yellow leaf curl China virus (TYLCCNV), the whitefly vector’s performance is promoted [39,81]. Furthermore, infection by tomato yellow leaf curl virus (TYLCV) in tomatoes disrupts JA signaling, resulting in the suppression of plant defenses and, consequently, an improved performance of its whitefly vector [39,81].

Some economically significant diseases transmitted by whitefly vectors include cotton leaf curl disease, cassava mosaic disease, and TYLC [82,83]. About 17 case studies of different begomoviruses have been documented for the plant-mediated effects of virus infection on non-viruliferous whitefly preference, and a total of 12 showed the host preference of whiteflies to the virus-infected plants [45,84,85,86,87], three indicated a whitefly preference for uninfected plants, [66,85,88] and two indicated no preference (Table 2) [31,69].

Tomato yellow leaf curl virus (TYLCV) is one of the most extensively researched begomoviruses, particularly concerning its effect on the preference of whiteflies without the virus. Several studies have reported that whiteflies tend to choose TYLCV-infected plants over uninfected ones [86]. According to Zhang et al. [93] both types of viruses (persistent and non-persistent) can impact the behavior and efficacy of insect vectors, either by altering the host’s phenotype or directly interacting with the insects to promote virus transmission. To date, researchers have conducted 19 case studies on the direct effects of begomovirus infection on whitefly preference to investigate whether viruliferous whiteflies exhibit a preference for virus-infected or uninfected plants [31,94]. Among these, 11 found that whiteflies preferred uninfected plants [69,87], two indicated a preference for virus-infected plants [69,86], and six showed no preference (Table 3) [45,69,95]. Begomoviruses transmitted by whitefly vectors have evolved to modify the preference of their vectors to increase their rate of transmission. Various studies have shown that this change in vector preference due to begomoviruses is conducive to transmission [79,96,97]. As a result, most studies on begomoviruses indicate that non-viruliferous whiteflies prefer virus-infected plants (Table 2), while viruliferous whiteflies prefer uninfected plants (Table 3). This preference manipulation of whitefly vectors directly promotes begomovirus transmission [31,50]. Begomoviruses have a genetically diverse population due to their error-prone replication processes. However, the duration of the virus infection may impact the plant-induced effect on whitefly preferences. Legarrea et al. [69] reported that there was no significant variation in the selection of non-viruliferous whiteflies (MEAM1 biotype) between TYLCV-infected and healthy tomato plants at three and twelve weeks after inoculation. However, a significant inclination towards TYLCV-infected plants was observed six weeks post-inoculation. Moreover, the manipulation of vector preference by the virus may vary depending on the whitefly species. Fang et al. [85] observed that MED whiteflies without the virus favored TYLCV-infected tomato plants for feeding, while MEAM1 whiteflies favored healthy plants.

Mixed infections with multiple viruses are prevalent [98,99], but the impact of mixed infections on the preference of whitefly remains poorly understood, and few studies have been described to date. Ban and his colleagues discovered that the non-viruliferous MEAM1 biotype of whiteflies favored feeding on TYLCV-infected tomato plants. However, they did not observe any preference between the single-viral-infected and mixed-infected plants [100]. Similarly, Gautam and his colleagues investigated the significant preference of both viruliferous and non-viruliferous whiteflies towards healthy plants instead of mixed- or single-viral infected plants [66]. Based on these findings, it can be concluded that mixed infections of whitefly and/or plants did not show any significant difference from single infections in terms of the whitefly preference.

The feeding behavior of whiteflies is critical for the transmission of begomoviruses, as these insects serve as vectors for numerous viral diseases affecting crops. The resulting impact on disease epidemiology highlights the importance of manipulating whitefly feeding behavior [31]. This manipulation carries ecological significance for all three contributors within this process: the plant, the whitefly, and the begomovirus. The change in whitefly feeding behavior could stem from a direct influence on whitefly physiology or indirectly through interactions with the plant. 

Seven articles have been published investigating the influence of virus infections on the feeding behavior of whiteflies through plant mediation [31]. The duration of whitefly sap-feeding is used as a standard index for collating results from various studies. Among the studies on viruses belonging to begomovirus, three have been reported for the plant-mediated effects of virus infection on whitefly feeding behavior, all of which found no effect [74,101]. Whitefly feeding behavior may be influenced by the acquisition of plant viruses, in addition to indirect effects induced by host plants. Two case studies on this subject have been reported. The first study compared the feeding behavior of whiteflies that carry the virus with those that do not, with both feeding on healthy host plants. Five cases showed an increase in whitefly feeding, while five showed no effect. The second research compared the feeding behavior of whiteflies with and without the virus on infected and healthy host plants and assessed its effects on both host plants and whiteflies. Three case studies have observed the plant-mediated effect of virus infection on whitefly feeding behavior. Interestingly, none of these cases revealed a noteworthy impact on it. Thus, the first scenario is termed the direct consequence of viral infection, whereas the second scenario involves the combined influence of both direct viral impact and plant-induced effects [31,72,75].

Regarding begomoviruses, five studies focusing on the direct effects of virus infection on whitefly feeding behavior have reported an increase in feeding on the infected host, while the other five suggest no effect [8,50,72,102]. Wang et al. [103] reported that TYLCV impacts the whitefly-feeding behavior resulting in increased secretion from salivary glands and uniform feeding on the healthy tomato plants. Similarly, Moreno-Delafuente et al. [50] observed the increased feeding frequency of whiteflies on egg plants infected with TYLCV. TYLCV has a direct effect on the behavior of whiteflies, while this effect is dependent upon the genotype of the host plants. He et al. [72] observed a positive mutual effect of TYLCV of a Chinese strain on MEAM1 whiteflies, but no direct effect of virus infection on whitefly feeding was observed [31]. Therefore, it is essential to note that a virus can have a variety of effects on whitefly behavior, even within the same pathosystem.

To manipulate the behavior of whiteflies through plant viruses, outcomes may differ depending on the specific species or strain/cultivar involved in the whitefly–virus–plant combination. For instance, nonviruliferous MEAM1 species prefer to feed on the healthy tomato plants, while MED tends to feed on infected plants [85]. On the other hand, the viruliferous MEAM1 biotype showed a feeding preference towards TYLCV-infected tomatoes of the Florida-47 cultivar [69]. Such variation indicates that changes in the behavior of whiteflies by viruses comprise a complex process that requires extensive careful experimentation for each unique pathosystem. Additionally, the careful selection of experimental materials is crucial to offer a reference for analyzing virus epidemics. Although several begomovirus species and whitefly species are present, only one or a few plant–virus–whitefly combinations has been used to determine the outcome of interactions on whitefly-transmitted plant viral epidemics. Therefore, to understand viral epidemics through the study of tripartite interactions, it is vital to first determine the involvement of each whitefly, virus, and plant species and then use them for investigation. A comprehensive examination of factors involving whitefly–plant–virus interactions is needed.

On the whitefly side of these interactions, MEAM1 individuals feeding on virus-infected plants showed the downregulation of genes involved in the oxidative phosphorylation pathway and detoxification enzymes. This reduced detoxification activity probably lowered energy costs and boosted whitefly performance [104]. However, such positive interactions could be pathosystem-specific. For instance, Gautam et al. [54] report that both *B. tabaci* MEAM1 and MED had significantly higher fitness advantage when feeding on TYLCV- and CuLCrV-infected plants compared to MEAM1 and MED individuals feeding on sida golden mosaic virus (SiGMV)-infected plants. Also, a transmission study from the same report concluded that *B. tabaci* MEAM1 efficiently transmitted TYLCV, CuLCrV, or SiGMV, whereas *B. tabaci* MED only transmitted TYLCV [54]. A subsequent follow-up gene expression study revealed no consistent pattern of gene expression in *B. tabaci* MEAM1 and MED upon the acquisition of TYLCV, CuLCrV, or SiGMV [105]. Taken together, these studies suggest a positive effect of begomovirus infection that can be perceived by non-vector *B. tabaci* cryptic species, and two highly similar viruses (CuLCrV and SiGMV, Bi-partite New World begomovirus) can interact very differently with *B. tabaci*. Thus, this signifies the context-specificity in begomovirus–whitefly interactions. Further studies are warranted to fully comprehend the events and factors that lead to positive effects of begomovirus in some systems and negative or no effects in others. 

Currently, there is a lack of comprehensive case studies, making it challenging to conclusively determine the effects of viruses on whitefly biology and feeding behavior when feeding on infected plants. Typically, the modification of vector effectiveness is attributed to changes in the synthesis of phytohormones or their signaling in plants [81,106,107]. Such studies could provide insights into the mechanisms of manipulation/changes in whitefly performance. 

This diversity allows for the rapid adaptive evolution of viruses since they may already possess gain-of-function mutations that are useful for different selection pressures [108]. Previous studies have demonstrated that even highly similar TYLCV variants could differentially influence component (whitefly–variant–plant) interactions and viral strain dispersal [109]. Therefore, strains of begomoviruses that are able to effectively manipulate whitefly preference to increase their transmission may gain a considerable edge over other viral entities and ultimately outcompete them. Taken together, over time, the genetic makeup of begomoviruses transmitted by whiteflies can experience modifications due to the continual selection pressure linked to whitefly transmission.

## 4. Begomovirus-Induced Changes in Plants

Begomoviruses have been documented to infect diverse crop species in tropical, subtropical, and temperate regions. The discovery of new species of these viruses has been facilitated by recent advancements in small RNA-based deep sequencing technology [110]. Our understanding of the natural host range of these viruses is expanding, and it has become evident that they not only infect herbaceous plants but also woody plants, such as citrus, grapevine, mulberry, and apple trees. The most common symptoms of various begomovirus attacks in plants include the yellowing of leaves, leaf curling and distortion, stunted growth, mosaic patterns on leaves, deformed fruits, necrosis, and dieback [111].

The virus can interfere with the plant’s immune system, preventing the plant from recognizing and responding to the virus infection. This allows the virus to replicate and spread throughout the plant. Begomoviruses can also cause changes in the expression of host plant genes, which can impact the plant’s physiological processes, including photosynthesis and hormone signaling. The virus may also induce the formation of abnormal structures in infected plants, such as leaf-like tissues on stems or the proliferation of roots. Symptoms of begomovirus infection differ based on various factors, such as the growth stage during initial infection, host species/cultivar, and begomovirus species, as well as weather conditions [67,112]. Several diseases may be indicated by green to bright yellow mosaic symptoms, leaf deformation, and chlorosis. Most begomoviruses are typically found in higher concentrations in newly expanded young leaves located in the uppermost sections of the plant, as opposed to older plant parts. Early infection in plants causes severe growth reduction and stunting, with reduced flowering and aborted fruit development leading to crop loss. However, later phases of development frequently experience milder illnesses and more bearable losses. Infected plants in some path systems can attract non-viruliferous vectors to acquire the virus while repelling viruliferous vectors, thus transmitting the virus to uninfected plants.

Several species of *B. tabaci*, which may or may not be carriers of plant viruses, can trigger modifications in the release of volatile organic compounds (VOCs) when interacting with different plant species. These changes in the volatile profile can either deter or attract other whiteflies [48,86,87]. Results indicate that infection with either virus, as well as infestation by whiteflies in the absence of viruses, led to both shared and distinct VOC accumulation. The unique VOCs detected in response to virus infection or insect infestation suggest that these triggers can elicit a specific response from plants. These VOCs could be used individually or in combination as a means of monitoring and disrupting pest populations [113,114]. Begomoviruses can manipulate plant-produced VOCs upon infection, which play a critical role in communication with insects [115]. Phytohormones, such as jasmonic acid (JA) and salicylic acid (SA), are crucial in responding to *B. tabaci* infestation [63,64,65]. Zhang et al. found that the infection of tobacco plants with Tomato yellow leaf curl China virus (TYLCCNV) suppresses JA-associated defenses, which favors the performance of the MEAM1 whitefly on virus-infected plants [93]. Similarly virus-infected plants suppress the synthesis and release of terpenoids, making them more favorable for whitefly performance than uninfected plants. In addition, plants infected with persistently transmitted viruses from the genus *Begomovirus* exhibit modified VOC emission profiles [78,116,117]. These alterations in VOCs can affect the behavior and performance of the insect vectors associated with these plants.

Host plant resistance is a highly effective method for managing diseases, particularly those caused by begomovirus, over a specific period [118]. In plant disease resistance, resistance (R) genes are responsible for triggering downstream signaling responses. These genes can be divided into several super-families based on their structural motifs, such as leucine-rich repeat domains, coiled-coil domains, interleukin-1 receptor domains, transmembrane regions, and nucleotide-binding-site-resistant proteins. Nucleotide-binding site leucine-rich-repeat (NBS-LRR) genes are the two most dominant R-genes in plants, and they are further sub-grouped based on their N-terminal coiled-coil or toll/interleukin-1 receptor (TIR) domain [119]. For instance, some of the important begomovirus resistance genes that confer resistance to TYLCV in tomatoes include Ty-1, Ty-2, Ty-3, and Ty-6. The Ty-1 gene uses RNA interference to combat TYLCV, while the Ty-2, Ty-3, and Ty-6 genes serve as receptors that can detect specific viral proteins. These receptor genes trigger defensive responses that block TYLCV infection. These genetic resistance mechanisms form a versatile defense strategy against begomovirus infection in tomato plants [120,121].

Plants have implemented various defense systems to protect themselves from viral infections. Since viruses are obligate intracellular parasites, they exclusively rely on the host’s machinery for replication and mobility [113,122,123,124,125]. RNA silencing is a fundamental gene regulation mechanism that specifically targets viral RNA, serving as one of the primary defense systems against viral infections in plants [123,126]. RNA silencing involves several steps, including silencing initiation, an effector phase, and an amplification phase, and it is regulated by RNA-dependent RNA polymerases (RDRs). Moreover, small interfering (siRNAs) act as mobile silencing signals that trigger local and systemic silencing upon their movement through phloem tissues [123]. However, to successfully infect the host, plant viruses encode proteins, known as viral suppressors of RNA silencing (VSRs), which counteract the antiviral RNA-silencing machinery [127]. Apart from suppressing RNA silencing, VSRs also play other roles during viral infection, including symptom induction, replication, and cell-to-cell movement. Therefore, it is of paramount importance to explore novel RNA silencing suppressors and gain a comprehensive understanding of their interactions with the plant’s RNA-silencing machinery. Several studies have been conducted on VSRs, including the identification of their structures and functions, but there is still much to learn about these proteins and their roles in viral infections. Geminiviruses encode various proteins, such as *C2*, *C4/AC4*, *V2/AV2*, *Rep*, and *C1* (encoded by the begomovirus-associated beta satellite), which have been reported to suppress RNA silencing at the post-transcriptional level (PTGS) and/or at the transcriptional level (TGS) in different ways [17,128].

In plants, gene silencing occurs through two mechanisms: PTGS and TGS [128,129,130]. TGS is a process that occurs within the nucleus and hinders gene expression by blocking the promoter region, thus preventing the binding of transcriptional machinery. Various methods contribute to TGS, including RNA-directed DNA methylation (RdDM), genomic imprinting, paramutation, transposon silencing, transgene silencing, and position effects. TGS is primarily responsible for silencing transposons and transgenes, while PTGS plays a secondary role in this regulatory process. In contrast, PTGS takes place in the cytoplasm and specifically targets and degrades the mRNA transcripts of specific genes. The methods utilized for PTGS encompass RNA interference (RNAi), clustered regularly interspaced short palindromic repeats (CRISPR/Cas9), and nonsense-mediated mRNA decay (NMD). Recent research has extensively explored gene silencing techniques, such as RNAi, virus-induced gene silencing, and CRISPR/Cas9 to bolster plant resistance against pathogens, improve drought tolerance, and engineer the ligno-cellulosic pathway. Within plants, small RNAs (sRNAs), like microRNA (miRNA) and small interfering RNA (siRNA), play a pivotal role in their defense against pathogens [131,132], while it is commonly believed that plant defenses mediate whitefly–begomovirus interactions [94,133,134]. 

The nutritional content of plants infected with begomoviruses may influence their suitability for whiteflies. In one study, African cassava *B. tabaci* from Uganda had a higher growth rate on East African cassava mosaic virus-Uganda (EACMV-UG)-infected plants, which correlated with a significant increase in the concentration of four amino acids in the phloem sap of virus-infected cassava plants. This suggests that the favorable interaction between whiteflies and EACMV-UG may be facilitated by improved nutrition in virus-infected plants. In contrast, the MEAM1 whitefly performed better on tobacco plants with TYLCCNV infection than on plants without infection [135]. However, no significant differences were found in the amino acid profiles, percentage of essential amino acids, or sugar-to-amino-acid ratio between infected and uninfected tobacco plants.

Managing and controlling begomovirus infections has become increasingly challenging due to the rapid evolution of new viral strains. Despite significant advancements in control strategies, such as RNAi, transgenics, and markers assisting breeding for host-resistant (R) genes, begomovirus remains a major problem in commercial crops. Begomoviruses have a high mutation rate of 10^−4^ substitutions per site per year (subs/site/year), which if accompanied by selection pressure imposed by the control strategies mentioned above, and in most cases can lead to the evolution of resistant breaking isolates of the viruses [136]. 

The relationship between begomoviruses and whitefly vectors is often indirect mutualism, as the whiteflies tend to perform better on virus-infected plants, which promotes virus spread and has significant agronomic effects. However, there is a lack of an understanding regarding how plant viruses manipulate plant defenses to promote vector performance. Recent studies have found that the C2 protein of begomoviruses lacking DNA satellites suppresses plant defenses against whitefly vectors, as demonstrated in the case of whitefly *B. tabaci* and tobacco. Specifically, the C2 protein of TYLCV, a devastating begomovirus that affects crops worldwide, interacts with plant ubiquitin to impede the breakdown of the JAZ1 protein. This interaction effectively hampers the JA defense pathway and suppresses the activity of terpene synthase genes regulated by MYC2; as a result, the enhanced survival and reproductive capabilities of whitefly vectors are facilitated [137]. These findings suggest that the inhibition of ubiquitination by begomovirus C2 protein may be a general mechanism in begomovirus, whitefly, and plant interactions. The interaction between geminiviruses and insect vectors has significant implications for the distribution and population dynamics of the vectors, as well as the epidemiology of begomovirus diseases.

## 5. Viral Adaptation and Genetic Changes during Plant–Virus–Vector Interactions

The understanding of genetic diversity in viruses is crucial for comprehending virus epidemiology and evolution. While the impact of genetic variation on virus–host interactions is well-established, its influence on virus–insect vector interactions is less understood. Mutation, recombination, and reassortment are the key drivers of genetic variation in viruses [138]. Before successfully infecting cells in new hosts, viruses must overcome the defensive responses of the host. To resist or tolerate infection, plants have evolved diverse mechanisms that restrict the replication and movement of viruses [139,140,141,142,143,144,145]. Concurrently, viruses have developed counteractive tactics to evade or suppress the host’s defense mechanisms [137,146]. This perpetual battle between plant defense mechanisms and viruses stimulates a co-evolutionary dynamic, influencing both entities involved.

Virus evolution is a dynamic process that leads to alterations in the genetic composition of viral populations over time. This process gives rise to novel viral variations, strains, and species, each possessing unique biological characteristics, including the ability to adapt to diverse host environments. This evolutionary process is influenced by various factors, such as hosts, vectors, the environment, and the viruses themselves [147,148]. The evolutionary trajectory of begomoviruses is influenced by the interplay of genetic drift and selection pressures stemming from host and vector factors. Within the viral population, random genetic variations or those that provide a competitive edge gradually permeate, giving rise to the emergence of new viral strains or species with unique biological characteristics.

While whiteflies transmit begomoviruses to plants, the virus particles have the potential to undergo mutations or recombination events with other related viruses present in the whitefly population [149,150]. These changes can alter virus properties, such as the host range, virulence, or transmissibility. Whiteflies may also selectively transmit certain virus variants, leading to changes in the overall genetic composition of the virus population and impacting its evolution and adaptation to different hosts. In conclusion, genetic variation is a key factor in the epidemiology and evolution of begomoviruses. Changes that occur during the transmission via whiteflies to plants can affect virus properties and influence their interactions with hosts and insect vectors. Further research is needed to fully understand the significance of genetic variation in virus–insect vector interactions. The process of recombination has led to the emergence of various recently identified species by combining genetic material from viruses that are either closely related or phylogenetically distant. Within the *Begomovirus* genus, notable examples include tomato yellow leaf curl sardinia virus, tomato yellow leaf curl Axarquia virus, sweet potato leaf curl Canary virus, tomato leaf curl Mahé virus, and tomato leaf curl Yunnan virus, all of which have originated as new species through recombination. 

Genetic variations within viral populations, genetic drift, and selection pressures from factors, such as the host, vector, and environment, collectively shape the evolution of viruses. Viruses, with their rapid replication rate exceeding that of their hosts, undergo constant changes to elude host immune responses while preserving their functionality across diverse hosts and vectors [151,152]. 

The relationship among plants, whiteflies, and begomoviruses is crucial for understanding the population dynamics and epidemiology of these organisms [153,154]. Several begomovirus species, including TYLCV associated with either the MEAM1 or Mediterranean cryptic species of the *B. tabaci* complex, have spread to different countries and regions [155,156,157]. In some cases, plant-mediated interactions between whiteflies and begomoviruses may have facilitated their introduction into new territories [94,135]. Plant defenses exert a dynamic effect on the adjustments and connections between vectors and pathogens. The modifications that begomoviruses undergo during transmission through whitefly vectors significantly contribute to boosting their genetic diversity, adaptability, and capacity to infect and proliferate across diverse host plants. Nevertheless, it is crucial to recognize that the precise changes observed might differ based on variables, such as the begomovirus strain, the whitefly vector species, and the specific host plant involved.

## 6. Conclusions and Future Prospects

Begomoviruses and whitefly species co-evolve as a result of genetic modifications in the virus genome, which call for intimate begomovirus–plant–whitefly interactions. Dynamic environmental conditions control these relationships. The manipulation issue can be confounding since some research demonstrates alterations in viruses and their vectors, while others claim that these changes may only be speculative. According to our perspective, begomoviruses affect the host and vector, respectively, making them more attractive to whiteflies and changing their feeding habits. To better understand begomovirus–plant–whitefly interactions, more research is needed because it is unclear whether these alterations are real or hypothetical. Plant virus epidemic prediction models can now be created because of recent technological advancements. Utilizing emerging technologies, like metagenomics, wisely and combining these contemporary tools with established techniques may help us better understand the intricate relationships between begomoviruses and whitefly vectors. This understanding may help us control the spread of begomoviruses and their whitefly vectors.

## Figures and Tables

**Figure 1 plants-12-03677-f001:**
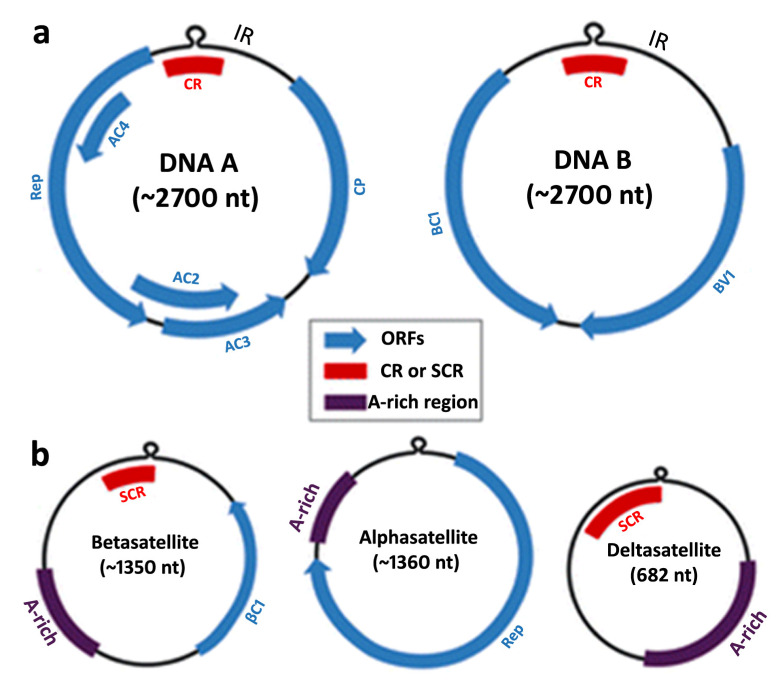
Begomovirus genome: (**a**) DNA-A segment of begomovirus, DNA-B segment found in bipartite begomovirus along with DNA-A. (**b**) Betasatellite, alphasatellite, and deltasatellite associated with monopartite begomoviruses.

**Figure 2 plants-12-03677-f002:**
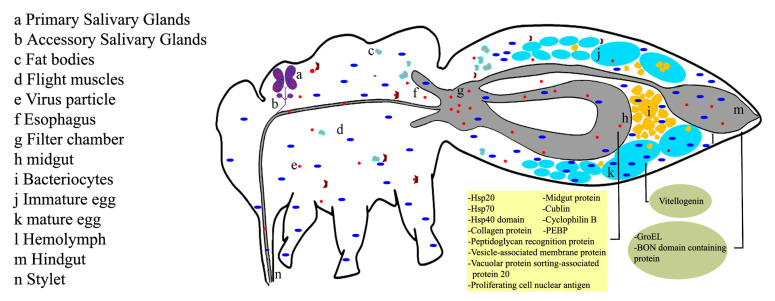
The intricate internal structure of the whitefly plays a crucial role in facilitating the circulative transmission of plant viruses by the *B. tabaci* species. The virus (e) is obtained through the ingestion of phloem sap and subsequently transported to the midgut (h) using the stylet (n) and esophagus (f). In the filter chamber area (g), the virus particles successfully breach the protective barrier of the midgut and gain entry into the hemolymph (l) by crossing the midgut plasmalemma and epithelial brush border. Once in the hemolymph, the virus circulates and ultimately reaches the primary salivary glands (a). Within these glands, the virus is internalized by passing through the basal lamina and secretory cells until it reaches the central lumen, which connects to the salivary gland duct. The virus then travels through the salivary gland duct and is expelled from the body during feeding via the salivary canal. It is important to note that the accessory glands (b) do not participate in the transmission process. Some virus particles that fail to enter the hemolymph are excreted from the body along with the honeydew through the hindgut. Additionally, the virus has the ability to invade developing oocytes (j) and eggs (k), potentially resulting in its transmission to the next generation through transovarial means. Furthermore, endosymbionts (i) play a significant role in transmission by secreting GroEL into the hemolymph.

**Table 1 plants-12-03677-t001:** Different viruses and modes of transmission by whiteflies.

Viruses	Mode of Transmission	Location	No. of Species	References
*Begomovirus*	Circulative, Non-propagative	Salivary glands	424	[32]
*Crinivirus*	Semi-Persistent	Foregut	14	[33]
*Ipomovirus*	Semi-Persistent	Unknown	7	[34]
*Torradovirus*	Semi-Persistent	Stylet	5	[35]
*Carlavirus*	Non-Persistent	Salivary glands	55	[36]
*Polerovirus*	Circulative, Non-propagative		2	[37]
*Cytrohabdovirus*	Unknown		1	[38]

**Table 2 plants-12-03677-t002:** Plant-mediated effects of virus infection on whitefly preference.

Host Plant		Virus-Isolate	Whitefly	Preference of Virus-Infected Versus Uninfected Plants by Non-Viruliferous Whiteflies	Reference
Common Name	Scientific Name				
Tobacco	*Nicotiana tabacum* cv. NC89	Tomato yellow leaf curl China virus-Y10	MEAM1	Virus-infected plants	[89]
Jimsonweed	*Datura stramonium* cv. unknown	Tomato yellow leaf curl virus-SH2	MED	Virus-infected plants	[90]
Tomato	*Solanum lycopersicum* cv. Zhongza9	Tomato yellow leaf curl virus-SH2	MED	Virus-infected plants	[85]
Tomato	*S. lycopersicum* cv. Florida 47	Tomato yellow leaf curl virus-unknown	MEAM1	Virus-infected plants	[69]
Tomato	*S. lycopersicum* cv. Security	Tomato yellow leaf curl virus-unknown	MEAM1	Virus-infected plants	[69]
Tomato	*S. lycopersicum* cv. Santa Clara	Tomato severe rugose virus-unknown	MEAM1	Virus-infected plants	[78]
Tomato	*S. lycopersicum* cv. Florida 47 R	Tomato yellow leaf curl virus-unknown	MEAM1	Virus-infected plants	[91]
Tomato	*S. lycopersicum* cv. Florida 47	Tomato yellow leaf curl virus-unknown	MEAM1	Virus-infected plants	[66]
Tomato	*S. lycopersicum* cv. Moneymaker	Tomato yellow leaf curl virus-SH2	MED	Virus-infected plants	[92]
Benthi	*N. benthamiana* cv. unknown	Tomato yellow leaf curl virus-SH2	MED	Virus-infected plants	[92]
Pepper	*Capsicum annum* cv. IIHR 3909	Chili leaf curl virus-unknown	MEAM1	Virus-infected plants	[87]
Tomato	*S. lycopersicum* cv. Moneymaker	Tomato yellow leaf curl virus-Israel	MED	Virus-infected plants	[86]
Tomato	*S. lycopersicum* cv. Zhongza9	Tomato yellow leaf curl virus-SH2	MEAM1	Uninfected plants	[85]
Tomato	*S. lycopersicum* cv. Santa Clara	Tomato severe rugose virus-unknown	MEAM1	Uninfected plants	[78,88]
Pumpkin	*Cucurbita pepo* cv. Goldstar	Cucurbit leaf crumplevirus-unknown	MEAM1	Uninfected plants	[66]
Tomato	*S. lycopersicum* cv. Florida 47	Tomato yellow leaf curl virus-unknown	MEAM1	No preference	[66,69]
Tomato	*S. lycopersicum* cv. Security	Tomato yellow leaf curl virus-unknown	MEAM1	No preference	[69]

**Table 3 plants-12-03677-t003:** Direct effects of virus infection on whitefly preference.

Host Plant		Virus-Isolate	Whitefly	Preference of Virus-Infected Versus Uninfected Plants by Viruliferous Whiteflies	Reference
Common Name	Scientific Name				
Tomato	*S. lycopersicum* cv. *Florida 47*	Tomato yellow leaf curl virus-unknown	MEAM1	Virus-infected plants	[89]
Tomato	*S. lycopersicum* cv. *Moneymaker*	Tomato yellow leaf curl virus-Israel	MED	Virus-infected plants	[90]
Cotton	*Gossypium hirsutum* cv. *F846*	Cotton leaf curl virus-unknown	Unknown	Uninfected plants	[85]
Cotton	*G. hirsutum* cv. *F846*	Cotton leaf curl virus-unknown	Unknown	Uninfected plants	[85]
Tomato	*S. lycopersicum* cv. *Florida 47*	Tomato yellow leaf curl virus-unknown	MEAM1	Uninfected plants	[69]
Tomato	*S. lycopersicum* cv. *Florida 47*	Tomato yellow leaf curl virus-unknown	MEAM1	Uninfected plants	[69]
Tomato	*S. lycopersicum* cv. *Security*	Tomato yellow leaf curl virus-unknown	MEAM1	Uninfected plants	[69]
Tomato	*S. lycopersicum* cv. *Security*	Tomato yellow leaf curl virus-unknown	MEAM1	Uninfected plants	[69]
Tomato	*S. lycopersicum* cv. *Santa Clara*	Tomato severe rugose virus-unknown	MEAM1	Uninfected plants	[78]
Tomato	*S. lycopersicum* cv. *Santa Clara*	Tomato severe rugose virus-unknown	MEAM1	Uninfected plants	[78]
Pumpkin	*C. pepo* cv. *Goldstar*	Cucurbit leaf crumple virus-unknown	MEAM1	Uninfected plants	[91]
Tomato	*S. lycopersicum* cv. *Florida 47*	Tomato yellow leaf curl virus-unknown	MEAM1	Uninfected plants	[91]
Pepper	*C. annum* cv. *IIHR 3909*	Chili leaf curl virus-unknown	MEAM1	Uninfected plants	[66]
Tomato	*G. hirsutum* cv. *F846*	Cotton leaf curl virus-unknown	Unknown	No preference	[92]
Tomato	*S. lycopersicum* cv. *Zhongza 9*	Tomato yellow leaf curl virus-SH2	MEAM1	No preference	[92]
Tomato	*S. lycopersicum* cv. *Zhongza 9*	Tomato yellow leaf curl virus-SH2	MED	No preference	[92]
Tomato	*S. lycopersicum* cv. *Security*	Tomato yellow leaf curl virus-unknown	MEAM1	No preference	[86]
Tomato	*S. lycopersicum* cv. *Moneymaker*	Tomato yellow leaf curl virus-SH2	MED	No preference	[85]
Benthi	*N. benthamiana* cv. *unknown*	Tomato yellow leaf curl virus-SH2	MED	Uninfected plants	[78,88]

## Data Availability

Not applicable.

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
