# Peer review of "A Review of Interactions between Plants and Whitefly-Transmitted Begomoviruses"

_plants, 2023, doi:10.3390/plants12213677_

Round 1

Reviewer 1 Report

The manuscript,” A review of interactions between plants and whitefly-transmitted begomoviruses” by Naveed et al illustrates the scientific knowledge related to begomovirus whitefly interactions. Although a number of reviews have been published on vector-plant virus interactions, this manuscript still has some new information. Therefore, it will be helpful for people working on vector virus interactions. However, the manuscript in its current form is not very attractive, a lot of stuff has been recirculated in the manuscript many times. I have highlighted some of them below. Also, I want authors to do more literature searches and include all recent papers on whitefly-begomovirus interactions. I have included some. Once these comments (see attachment) are addressed, we may have to take a fresh look at the manuscript. I recommend major revisions.  

Author Response

Dear Reviewer,

Thank you very much for your valuable comments and insightful suggestions. We strongly believe these inputs have significantly enhanced the quality of our manuscript. All the comments have been addressed and are provided in detail in the attached document, along with the track changes made in the original manuscript file.

Best regards

Reviewer 2 Report

This review deals with the interaction between begomoviruses, whiteflies vectors and host plants, focusing on different aspects of this interaction: the mechanisms of acquisition and transmission of begomoviruses, changes in the feeding habits suffered by infected whiteflies and physiological effects that begomovirus cause in infected plants. Finally, it deals with Viral Adaptation and Genetic Changes during plant-virus-vector interactions.

Important changes must be made so that the article can be published.

First, the word begomovirus is misspelled in several places. It should only be capitalized and italicized when it refers to the genus, as in line 396. In all other cases it should be lower case letters and not italicized.

Line 44 into two (There are three categories) primary categories concerning the duration of the vector's infection: persistent, semi-persistent, or non-persistent.

Line 54 while the others disperse in different ways as seen in Table 1 [13,14]. Table 1 is cited in the introduction but placed in the second section.

Lines 54 55 Begomoviruses have single-stranded circular DNA genomes which can exist in either mono or bipartite form and are usually transmitted by whiteflies [15-17]. This phrase does not fit in the context of the paragraph, it is repeated in the following paragraph.

Lines 59-58: Plants infected with the virus typically display a variety of symptoms such as leaf yellowing, leaf curling, stunted growth and reduced yield [11] Again, this phrase does not fit in the context of the paragraph.

Line 68, 72and 75 : The word Geminivirus is misused, since Geminivurus is a family that includes 9 genera, one of which is the  genus begomovirus, transmitted by whiteflies.

Line 75-76 There are over 1100 known types of plant pathogenic viruses, including geminiviruses that are transmitted by whiteflies These viruses are grouped into 9 genera.  this sentence is not clear. Maybe it will be better in this form There are over 1100 known types of plant pathogenic viruses, including geminiviruses. The family Geminiviridae  is grouped into 9 genera including Curtovirus, Begomovirus, Topocuvirus, Mastrevirus, Eragrovirus, Capulavirus, Becurtovirus, Turncurtovirus, and Grablovirus.

Line 89-90 while the DNA-B encodes two proteins responsible for facilitating the movement of viral DNA within and between host cells

Line 92 31]. . The intergenic region of DNA-A and, there are two points please remove one.

Lines 140 to 142: Once the whitefly feeds on the sap of a healthy plant, the virus can spread to it. The whitefly injects saliva into the plant while probing the tissue with its stylets, and the virus particles enter the plant cells along with the saliva seems to repeat the same concept in different words.

Lines 169 171: The electrical penetration graph (EPG) is a system widely employed to investigate the ingestion, acquisition, inoculation, and retention of vector-transmitted plant pathogenic viruses, both circulative and non-circulative [11,33,38] These lines do not make sense if some results are not discussed regarding transmission by whiteflies.

Lines 172-174 The interaction of several begomoviruses with their insect vectors has been observed to elicit various cellular, molecular, and behavioral responses that could impact vector survival and virus spread. it would be necessary to go deeper and give examples on the subject.

Lines 196-198 The viral infection in plants or whiteflies could have little or no effect on the feeding behavior of the whitefly. However, begomoviruses can have a direct or indirect impact on the behavior, fitness, and life cycle of insect vectors. in my opinion there is a contradiction between the first line and the last one.

Line 203 The behavior of whiteflies, like feeding and reference, can be influenced by viruses It is not clear what reference refers to in the context of the sentence.

Lines 228-229 Moreover, these viruses can also modify the production and activity of phytohormones, which affect the abundance of insect vector It would be interesting to include some examples of modifications caused by begomoviruses.

Lines230-232 These whitefly biotypes cause harm to the crops by directly feeding on them and also by facilitating the spread of economically important diseases such as cotton leaf curl di ease, cassava mosaic and TYLC. The previous concept was already reflected in paragraph 2 of the introduction only examples have been added.

Lines 232-234 About 17 case studies of different begomoviruses have been documented for plant-mediated effects of virus infection on Non-Viruliferous Whiteflies preference, and a total of 12 showed the host preference.

 Lines 296 303 Two case studies on this subject have been reported. The first study compared the feeding behavior of whiteflies that carry the virus with those that do not, with both feeding on healthy host plants. The second research compared the feeding behavior of whiteflies carrying and without the virus on infected and healthy host plants and assessed its effects on both host plants and whiteflies. Thus, the first scenario is termed the direct consequence of viral infection, whereas the second scenario involves the combined influence of both direct viral impact and plant-induced effects Explains two experiments but does not give the obtained results.

Lines 318-319 For instance, nonviruliferous MEAM1 biotype prefers to feed on the healthy tomato plants while MED biotype (MEAM1 is a cryptic species not a biotype)

Lines 399-407 Describes the different resistance genes but does not give examples of begomovirus resistance genes.

Lines 444-449: Zhang et al. found that infection of tobacco plants with Tomato yellow leaf curl China virus (TY-CCNV) suppressed JA-associated defenses, which favored the performance of the MEAM1 whitefly on virus-infected plants [81]. Similarly virus-infected plants suppressed the synthesis and release of terpenoids, making them more favorable to whitefly performance than uninfected plants.  I Thinj that this phrase is not related to RNA interference it could be included in paragraph 3

Line 531: strains of the B. tabaci complex Cryptic species not strains

In general it is well written, although sometimes there are phrases that are repeated and the concept, in these cases, is difficult to understand

Author Response

(The authors gave the same response as above.)

Round 2

Reviewer 1 Report

This version is far better than the last one. I have minor comments in the attachment. 

Author Response

Dear Reviewer,

Many thanks for the suggestion for improving the manuscript. First of all, I apologize, as I made all the changes and revised the manuscript according to your suggestions, but due to some unknown errors, many changes didn't appear in the PDF. Anyhow, I followed your recent suggestion and made corrections. I also rechecked the updated MS file and PDF version of the manuscript, and I hope you will find them correct this time.

Best Regrads,

Reviewer 2 Report

Lines38 39 Among hemipterans, Whiteflies are the most important group of vectors. a group  Whiteflies are a group of insects that consist of more than 1500 species, belonging to 161 genera across the world [6,7]

Maybe it is better: There are more than 1500 species of whiteflies belonging to 161 genera across the world

Line 59 The family Geminiviridae is grouped into 9 genera including, Geminiviridae with italic

Lines 92 93 Begomoviruses have single-stranded circular DNA genomes which can exist in either mono or bipartite form, and are usually transmitted by whiteflies .

This sentence can be omitted, I suggest starting directly When whiteflies feed on begomovirus infected plants

 Line 156 Genetically, these biotypes are divergent and reproductively isolated

Change biotypes to Cryptic species.

Lines 181- 183 The whitefly injects saliva into the plant while probing the tissue with its stylets, and the virus particles enter the plant cells along with the saliva. Inside the plant cells, the virus replicates and spreads to other parts of the plant, causing yellowing of leaf, stunted growth and decreased yield. Is a repetition of lines 175-177, I don`t know if it is deleted.

Lines 215-217 The interaction of several begomoviruses with their insect vectors has been observed to elicit various cellular, molecular, and behavioral responses that could impact vector survival and virus spread [42,55]. I couldn’t find the included examples as the authors said they added.

Lines 506-507 Some of the important begomovirus resistance genes include Ty-1, Ty-2, Ty-3, and Ty-6. [116,117] These genes seem to confer resistance to ToYLCV in tomato, but there is no explanation for this. Furthermore, the authors do not indicate how these genes act.

Minor editing of English language required

Author Response

(The authors gave the same response as above.)
